# Analysis of Maximization Strategy Intangible Assets through the Speed of Innovation on Knowledge-Driven Business Performance Improvement

**I Gusti Ayu Purnamawati** [1,*], **Ferry Jie** [2], **Puah Chin Hong** [3] **and Gede Adi Yuniarta** [1]

1 Faculty of Economics, Universitas Pendidikan Ganesha, Singaraja 81116, Indonesia; gdadi_ak@yahoo.co.id
2 School of Business and Law, Edith Cowan University, Joondalup, WA 6027, Australia; f.jie@ecu.edu.au
3 Faculty of Economics and Business, Universiti Malaysia Sarawak, Kota Samarahan 94300, Malaysia; chpuah@unimas.my
* Correspondence: ayu.purnamawati@undiksha.ac.id

**Abstract:** This study aims to determine the effect of human capital, structural capital, and consumer capital on financial performance and moderate the speed of innovation. The type of research used in the study is quantitative—data collection techniques in distributing questionnaires measured using a Likert scale. The sampling technique used was random sampling and was determined by the slovin formula. The population in this study was MSMEs in Buleleng Regency, and the samples used in this study amounted to 392 MSMEs. Data or statistical analysis techniques in the study were considered using the Structural Equation Model with WarpPLS 5.0 software modelling. The results show that the technology and commitment variables have no significant effect on the development of religious ecotourism villages. In contrast, cultural changes significantly impact the development of religious ecotourism villages. This study uses the speed of innovation as a moderating variable, the speed of innovation is one of the essential things for MSMEs to improve financial performance. The speed of innovation supports intellectual capital, which is currently focused on knowledge-driven business to create a competitive advantage.

**Keywords:** intellectual capital; thespeed of innovation; financial performance

## 1. Introduction

Financial performance can provide a tangible form in assessing whether a company is growing and developing well or not to determine the company's viability. Changes in the company's growth base from labor-based to knowledge-based lead to knowledge management that improves performance (Suseno and Pinnington 2017). Companies compete by running a knowledge-based business to use the available resources efficiently and economically.

Almost all companies now in their management develop knowledge-based business performance, which considers current assets and fixed assets as not a differentiator for competitive advantage (Nikolaou 2019). Competitive advantage in today's companies can be seen based on the knowledge and skills possessed by employees. To attract investors, the market value becomes the dominant value higher than the book value. This causes intellectual capital to be the capital that is most considered an intangible asset but can produce quality financial performance (Osinski et al. 2017). This intangible asset is not seen in traditional financial reporting, which only measures financial assets in the short term (Zambon 2017). In large companies, intangible assets have been presented and become a measurement of reporting to stakeholders and evaluating internally. In knowledge-based business, Hashim et al. (2015) said that high-quality resources are a priority for developed countries.

These resources in the company are realized in intellectual capital; three capitals include human capital concerning the knowledge, skills, and experience each employee brings in carrying out the company's operational activities (Kianto et al. 2017). Structural capital covers non-human resources in an organization such as databases, organizational charts, procedures, administrative processes, strategies, and everything of high value to the organization (Hejazi et al. 2016). The third capital is consumer capital by having knowledge in marketing activities and then establishing relationships with consumers and being a determinant in increasing market value (Harmeling et al. 2017). Human capital describes what an employee brings into a company so that the company's market value increases; structural capital describes how employees are connected within the company and how and when the employee leaves the company. Consumer capital describes how the company relates to external stakeholders (Dženopoljac et al. 2016). So human capital includes professional competence, employee motivation, and ability in leadership.

In contrast, structural capital includes collaboration with internal parties, understanding of IT and management tools and optimally in corporate culture, and consumer capital refers to the relationship between partners, consumers, and suppliers. as well as investors who have an overall effect on intangible assets (Wataya and Shaw 2019). Management can define, classify, assign, report and manage intangible assets in the company. These assets are the basis for claiming rights to future benefits that affect the creation of quality market value.

Its development and growth require competitiveness to generate profits; it is stated that human capital is the most complex factor in developing competitive advantage (Ployhart et al. 2014). However, the total cost of labor in various companies is larger than the cost of maintaining production and operations. These costs are different for each company, whether related to industry, goods or services. According to (Fathi et al. 2013), market value in a company is created based on intellectual capital, namely human capital, structural capital, and consumer capitalism, where the efficiency of the measured value is on the company's intangible and tangible assets.

Human capital is a genetic inheritance that includes education, training and experience, and business life. When companies can treat employees as well as possible, it impacts profits, and of course, this capital supports structural capital and consumer capital (Dumay et al. 2020). Structural capital in the company is the knowledge capable of fulfilling the company's activity processes and can support a high level of intellectuality (Komnenic and Pokrajčić 2012). However, when an organization has poor procedures and systems, it cannot achieve optimal performance and its potential is not utilized optimally. (Aghamirian et al. 2015) Consumer capital becomes potential in organizations, whereas knowledge already exists in the organization in establishing relationships with consumers, suppliers, competitors and the government. The better the relationship, the more excellent the opportunity for companies to learn from consumers and suppliers (Maharani and Fuad 2020). These three capitals are indispensable in improving a company's performance.

The current performance of MSME companies for the government can support the Indonesian economy (Hadiyati 2015), but MSME actors still feel some obstacles. The obstacles included here are both financial and non-financial. Economic barriers include the still weak MSMEs in the availability of funds, limited knowledge in managing finances, the lack of a systematic approach to MSME funding, and the lack of information obtained in seeking bank credit or operating costs that are still too high (Irjayanti and Azis 2012). Meanwhile, non-financial barriers include a lack of knowledge of the production aspects of MSMEs. They still use traditional technology, lack maximum quality control and product competitiveness, have not maximized innovation and creativity, have not been able to keep up with environmental changes, and are still weak in identifying marketing targets (Jovanovski et al. 2019). The product or service being marketed is not yet targeted at the needs or wants of the market. In addition, the problem that often occurs is that products circulating from neighbouring countries have product prices that are much cheaper with the

same quality, causing heavy competitiveness for local MSMEs. Both the lack of knowledge and skills possessed by MSME actors make the financial performance of MSMEs less than optimal, where their business is not productive, inefficient or ineffective (Casalino et al. 2014). MSMEs must be able to develop their potential and build a competitive advantage. The maximum financial performance obtained by MSMEs is based on intellectual capital that can be appropriately managed and accompanied by the speed of innovation so that MSMEs can run sustainably.

The whole world has indeed felt the impact of the COVID-19 pandemic, including Indonesia where what was supposed the most was the weakening of people's purchasing power due to workers who lost a lot of work (Affandi et al. 2020). The perpetrators of MSMEs certainly felt a reasonably significant impact, and some even went bankrupt. It is a formidable challenge for MSME actors, who were initially able to survive, and must innovate quickly (Hamdani and Wirawan 2012). The speed of innovation for MSME actors is significant to advancing their business. After the pandemic, MSMEs must rise to improve the Indonesian economy (Reardon et al. 2021). Indonesia is currently the largest market in Southeast Asia for agriculture and consumer goods; even the value of online sales transactions or e-commerce, which is growing more and more, is becoming the largest in ASEAN.

## 2. Methodology

### 2.1. Financial Performance

The stakeholder theory by R. Edward Freeman in 1984 stated that the existence of a company is strongly influenced by stakeholder support for the company (Dawkins 2014). Stakeholder power is seen as control over company resources (Miller et al. 2014). The higher the authority and attention given by the company to stakeholders, the stronger the relationship between the two parties (Lentjušenkova and Lapina 2016). Stakeholders can also influence management but must still see how stakeholder control functions over the company's resources (Beringer et al. 2013; James 2013). The Resource-Based View theory by Wernerfelt in 1984 focuses on significant company resources and capabilities as the basis for competitiveness and operational performance (Leonidou et al. 2017; Tehseen and Sajilan 2016; Barney and Mackey 2016). To achieve a competitive advantage, companies can utilize and develop sources of company capital, one of which is intellectual capital, which means using strategic assets in the form of intangible assets (Wijaya and Suasih 2020). This means that this theory views the company as having different assets, capabilities, experience, and organizational culture to give the company a sustainable competitive advantage.

Financial performance as a series of operational activities that focus on finance within a certain period is reported in financial statements, including income statements, changes in capital, balance sheets, cash flows and notes to financial statements (Abuzayed 2012), (Przychodzen and Przychodzen 2015). Financial performance in its measurement looks at implementation, financing and strategic decisions (Churet and Eccles 2014). Analyzing the financial performance of a company aims to find weaknesses. It can determine reliable strengths in financial performance so that they can make critical decisions for the company in the future (Revelli and Viviani 2015). Financial performance measurement by analyzing financial statements helps evaluate performance to improve it. (Teeratansirikool et al. 2013) Achievement of performance depends on the management, where every decision taken positively or negatively impacts financial performance.

## 2.2. Human Capital

The characteristics of human capital that are considered the most important in creating a relevant performance in organizations are education, knowledge and experience (Cherkesova et al. 2016). Teaching, learning gained, and previous experiences lead to innovative and creative discoveries to support the progress of operational activities, and implementing structural aspects helps achieve quality and success. Human capital is an investment in a company by providing education and training for its employees to improve employees' knowledge, skills, and competencies to maximize the productivity and output of a company (Malaolu and Ogbuabor 2013; Wright et al. 2014). These resources are essential for economic productivity, and skills are enhanced through education and experience. Human capital is divided based on competencies, namely knowledge, skills, and talents possessed by employees and attitudes, namely the willingness of employees to use abilities in providing a company benefits (Iwamoto and Suzuki 2019).

Human capital is one of the intangible assets of the company where intelligence, ability, creativity and innovation, as well as the experience possessed by all employees, can support the sustainability of the company, so that human capital is declared to be the main stakeholder in the company (Nuryani et al. 2018). Human capital is part of intellectual capital where companies employ employees who prioritize mindsets rather than physical ones to deal with technological developments (Sima et al. 2020). Investment in improving employees is expected to have a positive impact both in the short and long term of the company. As a form of an intangible asset, human capital is expected to create future economic value in the company by increasing employee competence and organizational ability (Silva et al. 2019). This illustrates how the quality of a company's workforce, the best quality, is beneficial for investors to assess efficiency and predict future profitability and company productivity (Yusuf 2013).

## 2.3. Structural Capital

Structural capital is a concept or system created by employees but owned within the company's scope. In companies based on a combination of internal structures that can improve the development process, establish initiatives, and improve technology, the structural capital also increases (Ling 2013). Structural capital includes non-human resources within the company, which have databases, organizational structures, organizational culture, strategies or other high value. Structural capital functions as leverage in company growth based on rules, databases, and corporate culture (Gogan et al. 2015). Structural capital includes the things in the company to be organized and integrated; this capital becomes the supporting infrastructure for performing well (Paunović 2021). Structural capital is a unique approach to carrying out operational activities because other competitors can hardly imitate it. By investing in structural capital, companies can improve their work processes so that services and quality of production and operational activities can be more efficient and effective (Aramburu et al. 2013). Structural capital in the company can produce high quality and reduce operating costs to lead to successful operational performance (Matos et al. 2017). This capital can direct to put aside unnecessary things in the process of creating value, helping achieve employee productivity and increasing income.

## 2.4. Consumer Capital

Consumer capital grows based on trust, commitment, norms, and interactive relationships between companies and stakeholders (Fisher 2019). Consumer capital is a facility for exchanging knowledge, learning and cooperation so that they can find ways to solve problems. This capital allows companies to develop good relationships with partners or customers so that organizations can learn from the experiences of others (Archer-Brown and Kietzmann 2018). Without good consumer capital, a company has difficulty interacting with its partners. Consumer capital embodies how continuous communication is applied by companies in serving external parties to add market value. Consumer capital has market share characteristics, customer retention, and profits from consumers (Kirmaci 2012). Man-

aged consumer capital is expected to overcome the worst conditions because the company can find out who its customers are. Without consumer capital, it is impossible to achieve market value and performance. Consumer capital is linked to the company's ability to have quality relationships with its clients and form a company's external business network, such as communicating with consumers and suppliers and having a reputation and brand (Jalali et al. 2014; Bagher Taghieh et al. 2013). Consumer capital is a relationship that companies must foster in running a business that includes consumers and suppliers. Consumer capital is a relational value between companies and other people, including consumer satisfaction, durability, reputation, suppliers, investors, government and business networks and other stakeholders (Isanzu 2015).

### *2.5. Speed of Innovation*

Innovation in the economic aspect means introducing new goods or services that customers do not know about or unique qualities of an item, introducing new production methods, or opening new markets (Snyder et al. 2016). Innovation is the most crucial factor in MSMEs to improve operational performance (Kuncoro and Suriani 2018). These three things are significant to avoid failures such as inappropriate strategies, non-innovative product designs, and costs incurred for product innovations that are too high or unable to compete with others (Lendel and Varmus 2014). The speed of innovation is expressed as the time required by the company from the concept and process to offering the new product to the market (Wang and Wang 2012). The speed of innovation has a reason as an innovation strategy, namely the result of the speed of innovation making superior new products to affect the company's performance (Hecker and Ganter 2013). The speed of innovation provides a sustainable competitive advantage, and intense competition and rapid technological developments make the pressure to innovate faster. A company's speed in innovating is very much needed, especially MSMEs, where MSME actors expect their business to survive and develop (Akman and Yilmaz 2019). Innovation is the main supporting factor in a company's success by looking at the basic innovation dimensions, including products, processes, marketing, and organization. A clear understanding of which direction to take in realizing innovation significantly assists a company in prioritizing market strategies. The speed of innovation is essential in competitive competition (Casadesus-Masanell and Zhu 2013). The ability to develop and launch innovative products quickly before being preceded by competitors is the right step for the key to the success of a company competing in the advanced technology industry. According to (Ngari and Muiruri 2014), in his research, the application of financial innovation has a significant effect on financial performance.

## 3. Results

The outer model is used to test the validity and reliability of a research instrument (Hair et al. 2014).

### *3.1. The Effect of Human Capital on Improving Financial Performance*

In employees, there is human capital with the attitudes, competencies, knowledge and skills, innovation, creativity and experience needed in an organization or company to achieve a target (Campbell et al. 2012). Companies realize that when they invest in their employees, they can easily enjoy better operational performance, which indirectly leads to increased financial performance (Wang et al. 2014). (Joshi et al. 2013) states that the existence of human capital in a company provides benefits, namely the loss of the threat of opportunism, the creation of synergies, reducing uncertainty, control and protection of assets, and the ability to access and follow the latest technological developments (Bendickson and Chandler 2017).

Human capital is needed to increase capabilities and maintain a competitive advantage to support sustainable investment and a knowledge-based economy (Chatterjee 2016). This capital is one of the differentiating factors for performance between companies that can bring economic value to the company. Increasing human capital internally can be performed by providing training and development programs and providing high-level knowledge and skills in facing the job competition market (Scafarto and Dimitropoulos 2018). According to Ogunyomi and Bruning (2016), human capital refers more to education, training, and other developments in improving employees' skills, knowledge, values, and social assets, resulting in employee satisfaction and indirectly affecting the company's financial performance. Nimtrakoon (2015) also states that the impact of human capital efficiency consistently affects financial performance.

**Hypothesis 1 (H1).** *Human capital affects improving financial performance.*

### 3.2. The Effect of Structural Capital on Improving Financial Performance

Structural capital influences financial performance. High structural capital certainly impacts efficiency in carrying out the production process and can reduce production costs. It increases company profits that affect assets (Yong et al. 2020). Good organizational management certainly impacts the creation of competitive advantage by expanding the company's capabilities. The existence of a structure that supports employees to produce intellectual rules and corporate culture is applied to provide optimal business performance (Nejati 2016). Structural capital is a strategic intangible asset in an organization which includes: organizational culture, procedures, information systems, hardware and software, databases, patents, copyrights, etc. When the organization strongly supports structural capital by fostering a culture of innovation and organizational commitment and involving top to bottom management, it creates a performance advantage, which means structural capital creates corporate value (Felício et al. 2014). Research conducted by Babai et al. (2016) shows that structural capital shows a company's ability to fulfil company processes and structures in supporting employee efforts to produce operational performance, especially optimal financial performance, for example, the company's operational system, organizational culture, organizational structure or technology system. According to research conducted by Thiagarajan and Baul (2014), structural capital becomes a means of infrastructure for employees to work optimally. When employees have a high intellectual level but the company's system is terrible, intellectual capital and financial performance are not able to be achieved optimally.

**Hypothesis 2 (H2).** *Structural capital affects improving financial performance.*

### 3.3. The Effect of Consumer Capital on Improving Financial Performance

Consumer capital is stated to influence financial performance; where consumer capital gets better and more competitive, it will affect maximum sales and use capital more efficiently to improve company performance (Ranani and Bijani 2014). Increased sales on consumer capital is due to a harmonious relationship between the company and its partners, such as quality and reliable suppliers, loyal and consistently satisfied customers with the services provided by the company, as well as good relations between the company and the government and the surrounding community (Hashemnia et al. 2014). Consumer capital is also external because it consists of relationships with outside parties and a network based on company satisfaction and loyalty (Deniswara et al. 2019). This means that this capital is a company's ability to meet consumer demand. When the organizational structure is reasonable, skilled employees providing quality services improve performance (Adnan et al. 2013). In consumer capital, the organization must determine the control and where its business is and how many chains to build for various people. Expanding the organization's business value chain is essential and pays attention to costs, risk mitigation, managerial decisions and the exploitation of the economic scope that impacts performance (Wuttke

et al. 2013). The benefits of empowering consumer capital are improving coordination, having interaction with consumers and suppliers, having the opportunity to create an idea or product differentiation and having control.

**Hypothesis 3 (H3).** *Consumer capital affects improving financial performance.*

*3.4. Speed of Innovation Moderates the Effect of Human Capital on Improving Financial Performance*

When creating innovation, knowledge and skills are needed to impact the product on the relationship between customers and company competitors (Meihami and Meihami 2014; Budi 2019). This indirectly indicates that the role of human capital is significant and related to increasing the productivity of MSMEs. Innovation decisions are made quickly and accurately based on environmental and external analysis. As MSME actors can speed up innovation, human capital must be accompanied by training in developing their skills; besides, they can also add insight to their business from financial and non-financial aspects (Agostini et al. 2017). With sufficiently mature knowledge and skills, and experience in building a business, the possibility of failure can be avoided and become a competitive advantage in the target market (Omotayo 2015). The performance of MSMEs can provide optimal results. Research by Wang et al. (2018) and Leitner (2018) showed that the speed and quality of innovation fully mediate the impact of human capital on financial performance.

**Hypothesis 4 (H4).** *Speed of Innovation moderates the effect of human capital on improving financial performance.*

*3.5. Speed of Innovation Moderates the Effect of Structural Capital on Improving Financial Performance*

Structural capital reflects non-human resources involved in organization, such as values, business processes, and behavior patterns. This capital facilitates the creation of innovation with information retrieval, retrieval, storage, processing, and analysis using technology systems (Wang et al. 2016). It means that structural capital is used to incorporate knowledge in a company so that there are acceptable standards to avoid conflicts due to changes in the realization of innovation (Wang et al. 2014). Innovation planning and decision-making become more productive and efficient when the capital structure is effective. It makes it faster to satisfy customer needs based on new products or services developed by the company (Costa et al. 2014). The existence of structural capital makes it easy to increase acquisitions, share knowledge, and build corporate culture. In the end, it can support the speed of innovation, fulfill process facilities and collaboration in developing innovations to improve financial performance (Soo et al. 2017; Ndubisi et al. 2015; Hartono and Halim 2014; Lestari et al. 2020). The speed of innovation is successful in improving the financial performance of MSMEs. Optimization is inseparable from the system and structure built by MSME actors in their business. These efforts include: the use of sophisticated technology, keeping up with the rapidly changing environment, optimally applied organizational culture, and the ability to adapt to processes. Innovative productivity becomes added value in the world market, achieving status by producing quality goods or services.

**Hypothesis 5 (H5).** *Speed of Innovation moderates the effect of structural capital on improving financial performance.*

*3.6. Speed of Innovation Moderates the Effect of Consumer Capital on Improving Financial Performance*

To achieving satisfactory innovation results, a company utilizes internal resources, science and technology, and external stakeholders' capabilities. Consumer capital forms an atmosphere of communication and provides feedback so that it is faster to develop the innovation process (Ungerman et al. 2018). Consumer capital embodies the company's solidarity with stakeholders, thereby creating an opportunity. Companies involve customers or suppliers, thus encouraging continuous innovation (Kianto et al. 2017). Activities in developing new products, work processes and quality, and services in the future lead to better company financial performance. The speed of innovation is very important for MSMEs in introducing and marketing new products through communication and establishing relationships with customers (Al-Ansari et al. 2013). The relationship fostered with customers, especially loyal customers, is considered to have a significant impact on responding to the introduction of innovations created by MSME actors. Here, MSME actors take a communication approach while introducing new products to customers and then allowing them to respond; if there are still some shortcomings, they can be corrected immediately (Aksoy 2017). In addition, this can build a more comprehensive network of cooperation with the government or other business partners. MSMEs can move more efficiently and effectively in procuring resources, producing, and even marketing to take advantage of business opportunities with other parties (Distanont and Khongmalai 2020). It is the right step in making decisions to innovate to provide operational and financial performance for MSMEs.

**Hypothesis 6 (H6).** *Speed of Innovation moderates the effect of consumer capital on improving financial performance.*

The research used is a quantitative approach, and a particular population or sample was researched. The data collection technique was distributing questionnaires to measure five variables: human capital, structural capital, consumer capital, innovation speed, and financial performance. The questionnaire used a Likert scale from 1- to 5 points. Summary of sample presentation as presented in Table 1. The sampling technique used was a random sampling technique determined by the solving formula. The population was MSMEs in the Buleleng Regency, which were recorded at the Disdagprinkopumkm.go.id in 2020 with a total of 54,489 MSME actors with a margin of error of 5%, so the sample obtained was 401 MSME actors. However, only 392 samples were in this study because nine questionnaires were not returned. Data or statistical analysis techniques in the study were considered using the Structural Equation Model (SEM) with WarpPLS 5.0 software modelling. Variable measurements are summarized in the presentation of Table 2.

**Table 1.** Summary of sample presentation.

| Sample Criteria | Number of Observations |
| --- | --- |
| Total questionnaires distributed | 401 |
| Complete questionnaires that were not returned | (9) |
| Total returned questionnaires | 392 |
| Complete questionnaires that cannot be processed | 0 |
| Total questionnaires that can be processed | 392 |

**Table 2.** Variables Definition.

| Variable | Definition | Data Source |
|---|---|---|
| Improved Financial Performance | A company's financial condition in a certain period regarding aspects of fundraising and its distribution is presented in the financial statements (Frias-Aceituno et al. 2014) | Questionnaire |
| Human Capital | Value is added in the form of knowledge, expertise, abilities and skills possessed by humans and then makes humans capital or assets of an organization to achieve strategic goals (Pasban and Nojedeh 2016) | Questionnaire |
| Structural Capital | Supporting infrastructure, organizational databases, and organizational design structures that enable human capital to carry out its functions for better performance (Sumedrea 2013) | Questionnaire |
| Consumer Capital | The knowledge possessed in aspects of marketing channels and relationships that occur with outside parties such as customers, suppliers, communities and governments that develop in the organization through operational activities (Kim et al. 2012). | Questionnaire |
| Speed of Innovation | The pace of progress manifests a bold appearance in innovating and commercializing new products, which means the company can accelerate its operational activities in developing new products (Engel 2015). | Questionnaire |

Research Model:

$Y = \gamma_1 X + \gamma_2 M + \gamma_3 XM + \varepsilon$

Description:

Y: Endogenous variables;

X: Exogenous variable;

$\gamma$: The influence coefficient of the exogenous on the endogenous latent variable;

M: Moderating variables;

The criteria for this validity are met if the loading value is 0.5 to 0.6. Convergent validity in this study is based on Tables 3 and 4; it is known that the combined loadings and cross-loadings have met the criteria, so the validity is fulfilled.

**Table 3.** Convergent validity.

| Variable | Indicator | X1 | X2 | X3 | Z | Y | *p*-Value | Description |
|---|---|---|---|---|---|---|---|---|
| Human Capital (X1) | X1.1 | 0.631 * | | | | | <0.001 | Valid |
| | X1.2 | 0.663 * | | | | | <0.001 | Valid |
| | X1.3 | 0.677 * | | | > | | <0.001 | Valid |
| | X1.4 | 0.709 * | | | | | <0.001 | Valid |
| | X1.5 | 0.741 * | | | | | <0.001 | Valid |
| | X1.6 | | | | | | <0.001 | Valid |
| Structural Capital (X2) | X2.1 | | 0.973 * | | | | <0.001 | Valid |
| | X2.2 | | 0.992 * | | | | <0.001 | Valid |
| | X2.3 | | 0.978 * | | | | <0.001 | Valid |
| | X2.4 | | 0.981 * | | | | <0.001 | Valid |

**Table 3.** *Cont.*

| Variable | Indicator | X1 | X2 | X3 | Z | Y | *p*-Value | Description |
|---|---|---|---|---|---|---|---|---|
| Consumer Capital (X3) | X3.1 | | | 0.638 * | | | <0.001 | Valid |
| | X3.2 | | | 0.768 * | | | <0.001 | Valid |
| | X3.3 | | | 0.745 * | | | <0.001 | Valid |
| | X3.4 | | | 0.718 * | | | <0.001 | Valid |
| | X3.5 | | | 0.823 * | | | <0.001 | Valid |
| Speed of Innovation (Z) | >Z1 | | | | >0.832 * | | ><0.001 | >Valid |
| | Z2 | | | | 0.776 * | | <0.001 | Valid |
| | Z3 | | | | 0.807 * | | <0.001 | Valid |
| | Z4 | | | | 0.797 * | | <0.001 | Valid |
| Improvement of financial performance (Y) | Y1 | | | | | 0.646 * | <0.001 | Valid |
| | Y2 | | | | | 0.786 * | <0.001 | Valid |
| | Y3 | | | | | 0.844 * | <0.001 | Valid |
| | Y4 | | | | | 0.913 * | <0.001 | Valid |

Source: processed data (* the validity has fulfilled the criteria with the loading value being 0.5 to 0.6).

**Table 4.** Convergent validity.

| Z × X1 | | Z × X2 | | Z × X3 | | *p*-Value | Description |
|---|---|---|---|---|---|---|---|
| Z1 × X1.1 | 0.596 | Z1 × X2.1 | 0.977 | Z1 × X3.1 | 0.867 | <0.001 | Valid |
| Z1 × X1.2 | 0.876 | Z1 × X2.2 | 0.997 | Z1 × X3.2 | 0.852 | <0.001 | Valid |
| Z1 × X1.3 | 0.831 | Z1 × X2.3 | 0.977 | Z1 × X3.3 | 0.688 | <0.001 | Valid |
| Z1 × X1.4 | 0.829 | Z1 × X2.4 | 0.985 | Z1 × X3.4 | 0.716 | <0.001 | Valid |
| Z1 × X1.5 | 0.934 | Z2 × X2.1 | 0.978 | Z1 × X3.5 | 0.852 | <0.001 | Valid |
| Z1 × X1.6 | 0.792 | Z2 × X2.2 | 0.988 | Z2 × X3.1 | 0.838 | <0.001 | Valid |
| Z2 × X1.1 | 0.892 | Z2 × X2.3 | 0.972 | Z2 × X3.2 | 0.932 | <0.001 | Valid |
| Z2 × X1.2 | 0.704 | Z2 × X2.4 | 0.957 | Z2 × X3.3 | 0.760 | <0.001 | Valid |
| Z2 × X1.3 | 0.866 | Z3 × X2.1 | 0.984 | Z2 × X3.4 | 0.721 | <0.001 | Valid |
| Z2 × X1.4 | 0.823 | Z3 × X2.2 | 0.986 | Z2 × X3.5 | 0.827 | <0.001 | Valid |
| Z2 × X1.5 | 0.808 | Z3 × X2.3 | 0.991 | Z3 × X3.1 | 0.798 | <0.001 | Valid |
| Z2 × X1.6 | 0.780 | Z3 × X2.4 | 0.990 | Z3 × X3.2 | 0.943 | <0.001 | Valid |
| Z3 × X1.1 | 0.861 | Z4 × X2.1 | 0.962 | Z3 × X3.3 | 0.794 | <0.001 | Valid |
| Z3 × X1.2 | 0.932 | Z4 × X2.2 | 0.992 | Z3 × X3.4 | 0.728 | <0.001 | Valid |
| Z3 × X1.3 | 0.744 | Z4 × X2.3 | 0.953 | Z3 × X3.5 | 0.937 | <0.001 | Valid |
| Z3 × X1.4 | 0.897 | Z4 × X2.4 | 0.964 | Z4 × X3.1 | 0.868 | <0.001 | Valid |
| Z3 × X1.5 | 0.852 | | | Z4 × X3.2 | 0.837 | <0.001 | Valid |
| Z3 × X1.6 | 0.791 | | | Z4 × X3.3 | 0.730 | <0.001 | Valid |
| Z4 × X1.1 | 0.850 | | | Z4 × X3.4 | 0.779 | <0.001 | Valid |
| Z4 × X1.2 | 0.800 | | | Z4 × X3.5 | 0.953 | <0.001 | Valid |
| Z4 × X1.3 | 0.851 | | | | | <0.001 | Valid |
| Z4 × X1.4 | 0.795 | | | | | <0.001 | Valid |
| Z4 × X1.5 | 0.848 | | | | | <0.001 | Valid |
| Z4 × X1.6 | 0.844 | | | | | <0.001 | |

Source: processed data.

Table 5 shows that the AVE value of each variable from 392 respondents is greater than the correlation between latent variables in the same column. It indicates that it can accept discriminant validity. It shows that it can accept discriminant validity.

**Table 5.** Discriminant validity.

| Correlations among l. vs. with sq. rts. of AVEs | | | | | | | | |
|---|---|---|---|---|---|---|---|---|
| | **X1** | **X2** | **X3** | **Z** | **Y** | **Z × X1** | **Z × X2** | **Z × X3** |
| Human Capital (X1) | 0.634 * | | | | | | | |
| Structural Capital (X2) | | 0.902 * | | | | | | |
| Consumer Capital (X3) | | | 0.603 * | | | | | |
| Innovation Speed (Z) | | | | 0.739 * | | | | |
| Improvement of financial performance (Y) | | | | | 0.613 * | | | |
| Z × X1 | | | | | | 0.588 * | | |
| Z × X2 | | | | | | | 0.708 * | |
| Z × X3 | | | | | | | | 0.520 * |

Source: processed data. (* Cronbach's alpha value is greater than 0.5. All variables meet the reliability standards).

From Table 5, the composite reliability value of each variable is above 0.7, and the Cronbach's alpha value of each variable is above 0.5. It is concluded that all variables have met the reliability criteria. In addition, Table 6 shows the R-square in this study is 0.393, which means that 39.3% of the variables of increasing financial performance can be explained by the variables of human capital, structural capital, and consumer capital and the speed of innovation as moderating variables, while the remaining 60.7% are influenced by other variables. The standard method bias of the research results are worth less than 3.3, so the total collinearity value of VIFs is accepted. In the measurement of Q-square, coefficients are used as an assessment of predictive validity, which can be negative. They have a value greater than 0, and Table 6 shows that the value is more significant than 0, so it is declared valid.

**Table 6.** Latent variable coefficients.

| | **X1** | **X2** | **X3** | **Z** | **Y** | **Z × X1** | **Z × X2** | **Z × X3** |
|---|---|---|---|---|---|---|---|---|
| R-squared coefficients | | | | | 0.393 | | | |
| Adjusted R-squared coefficients | | | | | 0.311 | | | |
| Composite reliability coefficients | 0.798 | 0.946 | 0.732 | 0.828 | 0.693 | 0.876 | 0.941 | 0.806 |
| Cronbach's alpha coefficients | 0.696 | 0.923 | 0.547 | 0.722 | 0.513 | 0.854 | 0.933 | 0.749 |
| Average variances extracted | 0.401 | 0.813 | 0.364 | 0.547 | 0.363 | 0.238 | 0.501 | 0.177 |
| Full collinearity VIFs | 2.893 | 1.035 | 1.732 | 2.181 | 1.448 | 1.506 | 1.013 | 1.264 |
| Q-squared coefficients | | | | | 0.335 | | | |

Source: processed data.

The significance level of testing this hypothesis is performed by looking at the value of the *p*-value. (Davcik 2014) The inner model test or structural model evaluation is a specification in determining the relationship between latent constructs and other latent constructs. Research is declared good if the structural model meets the required standards (Kock 2015). In Table 7 below, there are test items and standard test values for the inner model used to measure the model's strength.

**Table 7.** Model fit and quality indices.

| No. | Model Fit and Quality Indices | Fit Criteria | Indeks | Description |
|---|---|---|---|---|
| 1 | Average path coefficient (APC) | $p < 0.05$ | 0.169 | Fulfilled |
| 2 | Average R-squared (ARS) | $p < 0.05$ | 0.311 | Fulfilled |
| 3 | Average adjusted R-squared (AARS) | $p < 0.05$ | 0.293 | Fulfilled |
| 4 | Average block VIF (AVIF) | acceptable if $\leq 5$, ideally $\leq 3.3$ | 1.341 | Fulfilled |
| 5 | Average full collinearity VIF (AFVIF) | acceptable if $\leq 5$, ideally $\leq 3.3$ | 1.634 | Fulfilled |
| 6 | Tenenhaus GoF (GoF) | small $\geq 0.1$, medium $\geq 0.25$, large $\geq 0.36$ | 0.364 | Fulfilled, Categori Large |
| 7 | Sympson's paradox ratio (SPR) | acceptable if $\geq 0.7$, ideally = 1 | 0.767 | Fulfilled |
| 8 | R-squared contribution ratio (RSCR) | acceptable if $\geq 0.9$, ideally = 1 | 0.951 | Fulfilled |
| 9 | Statistical suppression ratio (SSR) | acceptable if $\geq 0.7$ | 0.833 | Fulfilled |
| 10 | Nonlinear bivariate causality direction ratio (NLBCDR) | acceptable if $\geq 0.7$ | 0.833 | Fulfilled |

Source: processed data.

Based on the results of the output in Table 7, the fit and quality indices model for all criteria are known. The values of APC, ARS, AARS, AVIF, and AFVIF to GoF have met the requirements so that the structural model can be accepted and used for analysis.

Figure 1 shows a direct relationship for the variables studied where the output results are in the form of a model and the results of the path analysis test. Table 8 output results in path coefficient values are used to determine the magnitude of the influence of direct and indirect relationships (moderation). The results of the direct influence test in this study are shown in Figure 1 and Table 7: the path coefficient value of human capital, structural capital, and consumer capital, towards improving financial performance is 0.205 ($p$-values < 0.001), 0.157 ($p$-values 0.007 < 0.05), and 0.362 ($p$-values < 0.001), which means that all variables have a positive and significant effect on improving financial performance; The path coefficients of the speed of innovation to moderate human capital on improving financial performance are 0.054. The $p$-values of >0.202 with a significance level of 0.05 stated that the speed of innovation does not moderate the effect of human capital on improving financial performance. The path coefficients of the speed of innovation moderating structural capital on improving financial performance are 0.119 and $p$-values of 0.031 < 0.05 significance level—the speed of innovation moderates the effect of structural capital on improving financial performance. The value of path coefficients from the speed of innovation to moderate consumer capital on improving financial performance is −0.113 and $p$-values of 0.038 < 0.05 significance level. It is stated that the speed of innovation is not strong enough to moderate the effect of consumer capital on improving financial performance.

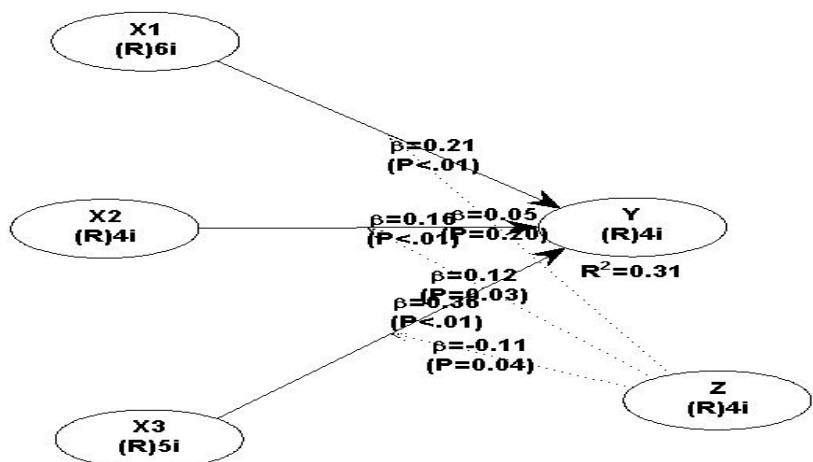

**Figure 1.** The analysis results of the direct and indirect effects.

**Table 8.** Path coefficients.

| Variable | Criteria | | Description |
|---|---|---|---|
| | **Path Coefficients** | ***p* Values** | |
| Human Capital (X1) | 0.205 | <0.001 *** | Highly Significant |
| Structural Capital (X2) | 0.157 | 0.007 *** | Highly Significant |
| Consumer Capital (X3) | 0.362 | <0.001 *** | Highly Significant |
| Innovation Speed× Human Capital (Z × X1) | 0.054 | 0.202 | Not significant |
| Innovation Speed× Structural Capital (Z × X2) | 0.119 | 0.031 ** | Significant |
| Innovation Speed× Consumer Capital (Z × X3) | −0.113 | 0.038 ** | Significant |

Source: processed data (** a *p*-value of 0.05 = significant; *** a *p*-value of 0.01 = highly significant).

## 4. Discussion

Intellectual capital, which has three components, is increasingly important for MSME actors to apply. They must follow increasingly sophisticated technological developments and difficult business competition, so MSME actors must change business patterns into knowledge-based businesses. MSMEs continue to strengthen the foundation and amount of employee talent they have to face the digital world so that the company does not experience economic downturns in the future. The use of tangible assets and intangible assets for MSME actors is expected to be more efficient in improving their business financial performance.

H1 shows that the first hypothesis is accepted. The results of this study are similar to research conducted by (Martin et al. 2013; Kim et al. 2016; Post and Byron 2015; Gambardella et al. 2013) who found that the higher the value of human capital, the higher the achievement of financial performance. Human capital is the component of intellectual capital that has the most significant role in an organization. When the company treats employees as human capital or assets, greater profits are obtained than when treated as resources. Human capital becomes a competitive and sustainable superior resource so that it becomes valuable capital, cannot be imitated and cannot be substituted. As stated by Ployhart (2012), human capital cannot be replicated because it is based on individuals. Individuals are the basis for forming organizational structures and systems that are difficult for other competitors to follow. Many companies make labor productivity a measure of the company's financial performance to see how the company has carried out the performance results in a period (Shaw et al. 2013). Human capital is an advantage in the organization

where it is extracted from the knowledge of an employee and becomes the most valuable asset (Crane and Bontis 2014).

Furthermore, the following study results indicate that H2 is accepted, where the results of this study are the same as the research of Alipour (2012) and Musibah and Alfattani (2014). Structural capital creates a work environment that helps MSME actors increase knowledge about the business they are running to improve their financial performance (Sardo et al. 2018). Structural capital includes the company's operational systems, production processes, organizational culture, management philosophy, and everything the company owns. Managing structural capital can develop knowledge, shorten the time of a job, and have a clear strategy (Sydler et al. 2014; Surya et al. 2021) This management produces something new and can be learned so that MSME actors have progressive and increasing productivity.

The results of the study indicate that H3 is accepted. As is the case in research by Baporikar et al. (2016), who found that consumer capital is one of the sources of income components in improving financial performance, one of which can be performed by retaining old customers and attracting the attention of new customers so that it becomes an essential aspect for MSMEs. MSME actors determine and then select the selected customers and market segments to target for their business. In consumer capital, how much value is given by consumers or customers can be seen from the products issued, good relations with customers, and brand image (Callarisa et al. 2012). A product is valuable if it approaches or even exceeds what consumers perceive. This creates potential consumers, so it is also necessary to provide the best service based on existing capabilities and resources (Teece 2018); this certainly impacts company performance, especially financial performance.

Furthermore, the results of the moderation research show that H4 is rejected, which means that the speed of innovation cannot moderate the influence of human capital on improving financial performance. It is inconsistent with research conducted by Aryanto et al. (2015) and Zhou et al. (2013), who found that human capital practices are positively related to innovation ability, which positively affects innovation performance. The speed of innovation does not only rely on human capital, but companies sometimes mostly prefer to use technology (Murray et al. 2016). One of the reasons is because technology is easier to apply when compared with human capital, which must be given education and training which requires more time and costs. In addition, innovation using human capital is more for small companies than large-scale companies (McGuirk et al. 2015). Human capital utilized efficiently is a determining factor in increasing productivity, but this is not entirely the case because it sometimes triggers waste and inefficiency, significantly so when speeding up innovation without sufficient knowledge and experience (Tzabbar and Margolis 2017). This reduces the financial performance of a company.

Further research indicates that H5 is accepted, which means that the speed of innovation can moderate the effect of structural capital on improving financial performance. The results of this study follow research conducted by Bayraktaroglu et al. (2019), who found that innovation has a moderating effect on the relationship between structural capital and profitability. Structural capital is an indicator of the added value of a company, which includes the database, organizational structure, corporate culture, strategy and other matters related to the company. Structural capital becomes a liaison for resources that have more value; this is because structural capital is a means and infrastructure in supporting the performance of MSME actors, especially in creating an innovation (Chahal and Bakshi 2015; Alrowwad and Abualoush 2020). When employees have high knowledge, but they are not supported by adequate facilities in applying the speed of innovation, they are not able to improve financial performance. Structural capital plays an essential role in all activities of MSME actors, especially in realizing innovations that enhance the financial performance of their businesses (Han and Li 2015).

Following the study results, H6 is rejected, which means that the speed of innovation is not strong enough to moderate the effect of consumer capital on improving financial performance. The results of this study follow research conducted by Ratnawati (2020),

who found that consumer capital does not always provide financial benefits for MSME actors. This is because customers do not all have a high level of loyalty, especially new customers (Li 2014); sometimes, these customers become a threat to SMEs, where they prefer to compare products with other competitors. Especially for the innovations created by MSME actors, sometimes having the speed of innovation can be an opportunity for other competitors to follow and even exceed the products that have been made (Mehdivand et al. 2012; Campo et al. 2014). So, here, it is related to which MSME actors can respond as quickly as possible but still follow the needs of outsiders, especially customers. So MSME actors who have the speed of innovation then take advantage of consumer capital, but it is not to the needs of outsiders; of course, this has an impact on the financial performance expected by MSME actors.

## 5. Conclusions

The study results, using five variables, show shown differences between the research results and the hypothesis in this study. Developments in Industry 4.0 have caused the business environment to become increasingly uncertain. It will also affect the decline in the national economy, so companies are expected to need to make quick and steady management decisions to maintain their sustainability and strengthen their position in the market supported by the government. Currently, most of the business world has adopted a knowledge-based business that emphasizes using the intellectual capital owned by the company, including human capital, structural capital, and consumer capital. Intellectual capital is also one of the important assets in the success and survival of MSMEs. For the government, micro, small, and medium enterprises are an essential part of the backbone of the Indonesian economy. To maintain the sustainability of MSMEs, a competitive advantage is needed by utilizing their intellectual capital. Intellectual capital is an intangible asset that helps create the best product and service innovations to increase the performance of MSMEs. The Resource-Based View (RBV) theory states that a competitive advantage is achieved if a company can use and maximize its resources. The RBV theory is considered appropriate to describe the company's internal strength, which is carried out through intellectual capital, which impacts financial performance. The limitations of this study are that the variables used are only three independent variables: human, structural capital, and consumer capital, as well as the moderating variable, namely the speed of innovation, so that it still does not show a role in improving financial performance. They represent MSMEs outside the Buleleng Regency area.

**Author Contributions:** Conceptualization, methodology, writing—original draft, I.G.A.P.; data curation, formal analysis, supervision, F.J.; writing—review and editing, P.C.H.; resources, investigation, visualization, G.A.Y. All authors have read and agreed to the published version of the manuscript.

**Funding:** This research received no external funding.

**Institutional Review Board Statement:** Not applicable.

**Informed Consent Statement:** Not applicable.

**Data Availability Statement:** Not applicable.

**Acknowledgments:** Thank you for the support provided by the Institute for Research and Community Service Ganesha University of Education, and the Ministry of Research Technology, National Innovation Research Agency, Republic of Indonesia.

**Conflicts of Interest:** The authors declare no conflict of interest.

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
