# Peer review of "Analysis of Maximization Strategy Intangible Assets through the Speed of Innovation on Knowledge-Driven Business Performance Improvement"

_economies, doi:10.3390/economies10060149_

Round 1

Reviewer 1 Report

  1. Bring the motivation and issue at hand. After reading the introduction, the main issue is missing from the important discussion. It is good to bring the localised issue in this study the MSME in Buleleng Regency into discussion. 
  2. Objectives are missing in the introduction.
  3. Section 2 and then jump to section 3. something missing here. Literature review more like it for section 2. 
  4. Why applied to Buleleng regency?
  5. What XM represent in the equation below Table 2?
  6. Moderating results?
  7. Table  8 - hypothesis? 
  8. Link back the results to the localised implication. How does the MSMEs benefit from this research? government intervention and policy for MSMEs?

Author Response

REVIEWER 1

  1. Introduction and discussion have been adapted to the research problem as follows.

Answer:

“Both large, medium, and MSME companies in carrying out their operational activities strive to have the most important asset, namely human resources which have a major influence on the success of a business. However, in various studies conducted, it is still found that the frequent and serious obstacle faced by MSMEs is the low quality of human resources, not least what happened to MSME actors in Buleleng Regency. Based on a study by the Balitbang Buleleng, this was felt during the COVID-19 pandemic where several MSME actors experienced losses and even went bankrupt because they did not improve the quality of their business souls, such as not carrying out innovation to maintain their business development. In addition, low education and lack of mastery of science and technology make it difficult for MSME actors to accept and adapt to the renewal process. This is the reason why the research was conducted in the Buleleng district. In addition, this research was conducted because previous research only examined the problems faced by MSMEs in the field of intellectual capital, so the researchers added several aspects to create research novelties and the results of previous studies were still diverse. Furthermore, the difference between this study and previous research lies in the research location, research subjects, and analytical techniques used. Based on the background and problems raised, this study aims to determine 1) the effect of human capital, structural capital, and consumer capital on improving financial performance, and 2) the speed of innovation as a moderating variable (strengthening or weakening) to determine its effect on the relationship between capital human capital, structural capital and consumer capital to improve financial performance” (found on line 114)

“Intellectual capital consisting of human capital, structural capital, and consumer capital is needed by MSME actors. Especially Buleleng district is facing competition and improving MSME financial performance. Things need to be considered regarding the guidance and development of human resources, organization and application of technology, and marketing and promotion. Meanwhile, the speed of innovation does not always have a positive impact on improving financial performance, because some things such as innovation are conveyed to customers too quickly so that they are not always well-received” (found on line 568)

  1. Goals have been laid out in the introduction as follows

Answer:

“Based on the background and problems raised, this study aims to determine 1) the effect of human capital, structural capital, and consumer capital on improving financial performance, and 2) the speed of innovation as a moderating variable (strengthening or weakening) to determine its effect on the relationship between capital human capital, structural capital and consumer capital to improve financial performance.” (found on line 129)

  1. The literature review section has been adjusted, which is located on lines 256, 279, 302, 323, 341, 365
  2. The reasons were conducted in Buleleng have been explained as follows:

Answer:

“The research was conducted in the Buleleng district considering that this district is the largest district in the province of Bali, in addition to the many potential resources and tourism places it has, many MSME actors take advantage of this and are able to develop it. This is evidenced by the fact that Buleleng Regency is ranked third with the highest number of MSMEs in Bali Province in 2020.” (found on line 398)

  1. XM has been described in X*M: Exogenous variables* Moderating Variables (found on line 415)
  2. Moderation has been changed; it is found on lines 338, 362, 388, 526, 540, 554
  3. The wording of the hypothesis has been changed to “research results obtained” (found on line 478)
  4. The implications have been adjusted to the following conclusions:

Answer:

"Following the research that has been done and the results obtained, the implications of this research on improving the financial performance of MSMEs where it is necessary to maximize the potential of human resources owned by MSMEs to increase competitive advantage. The results of this study are expected to be a reference and consideration related to intellectual capital issues and speed of innovation and how these things can affect the improvement of financial performance by MSMEs. Through this research, MSMEs are expected to pay more attention to and utilize all human capital, structural capital, and consumer capital to improve performance, especially financial performance. Not only intellectual capital but the speed of innovation must be maximized in order to determine the best competitive strategy that can be carried out by MSMEs in Buleleng Regency. The results of this study can be taken into consideration by the Buleleng district government in setting policies to encourage the development and improve the performance of MSMEs. One of which is to conduct a coaching and development program for MSME skills with the target of focusing on the intellectual capital owned by MSMEs in Buleleng Regency." (found on line 579)

Reviewer 2 Report

I am pleased to have the opportunity to review this research paper. Although the topic of this research study is interesting and fits within the journal scope, I think authors should apply the comments indicated below to increase the quality of research justification, contributions, and findings. The manuscript knows lacks in scientific style and structure.

What is the originality of this research?  Paper research gap and originality should be better presented at the end of introduction section

Please consider this structure for manuscript final part.

-Discussion

-Conclusion

-Managerial Implication

-Practical/Social Implications

Questions to be answered: What practical/professional and academic consequences will this study have for the future of scientific literature (theoretical contributions)?

Why is this study necessary? should make clear arguments to explain what the originality and value of the proposed model is. This should be stated in the final paragraphs of introduction and conclusion sections.

Good luck!

Author Response

REVIEWER 2

  1. The discussion section has been adjusted as follows:

“Intellectual capital consisting of human capital, structural capital, and consumer capital is needed by MSME actors. Especially Buleleng district is facing competition and improving MSME financial performance. Things need to be considered regarding the guidance and development of human resources, organization and application of technology, and marketing and promotion. Meanwhile, the speed of innovation does not always have a positive impact on improving financial performance, because some things such as innovation are conveyed to customers too quickly so that they are not always well-received” (found on line 568)

  1. The conclusion section is adjusted as follows:

“In accordance with the research that has been done and the results obtained, the implications of this research on improving the financial performance of MSMEs where it is necessary to maximize the potential of human resources owned by MSMEs so as to increase competitive advantage. The results of this study are expected to be a reference and consideration related to issues of intellectual capital and speed of innovation, and how these things can affect the improvement of financial performance by MSMEs. Through this research, MSMEs are expected to pay more attention to and utilize all human capital, structural capital, and consumer capital in order to improve performance, especially financial performance. Not only intellectual capital but the speed of innovation must be maximized in order to determine the best competitive strategy that can be carried out by MSMEs in Buleleng Regency. For the government, the results of this study can be taken into consideration by the Buleleng district government in setting policies to encourage the development and improve the performance of MSMEs, one of which is to conduct a coaching and development program for MSME skills with the target of focusing on the intellectual capital owned by MSMEs in Buleleng Regency. Developments in industry 4.0 have caused the business environment to become increasingly uncertain. It will also affect the decline in the national economy, so companies are expected to need to make quick and steady management decisions to maintain the company's sustainability and strengthen their position in the market supported by the government. Currently, most of the business world has adopted a knowledge-based business that emphasizes using the intellectual capital owned by the company, including human capital, structural capital, and consumer capital. The limitations of this study are that the variables used are only three independent variables: human, structural capital, and consumer capital, as well as the moderating variable, namely the speed of innovation so that it still does not show a role in improving financial performance. They are representing MSMEs outside the Buleleng Regency area. It is hoped that further research will be able to develop variables and research sites that are supporting in improving financial performance, and can use different analytical techniques so as to strengthen research results.” (found on line 579)

  1. Managerial implications have been made in this study:

“ In accordance with the research that has been done and the results obtained, the implications of this research on improving the financial performance of MSMEs where it is necessary to maximize the potential of human resources owned by MSMEs so as to increase competitive advantage. The results of this study are expected to be a reference and consideration related to issues of intellectual capital and speed of innovation, and how these things can affect the improvement of financial performance by MSMEs.” (found on line 579)

  1. Practical/social implications have been made in this research:

“Through this research, MSMEs are expected to pay more attention to and utilize all human capital, structural capital, and consumer capital to improve performance, especially financial performance.  To determine the best competitive strategy that MSMEs can carry out in Buleleng Regency, Not only intellectual capital but the speed of innovation must be maximized. The results of this study can be taken into consideration by the Buleleng district government in setting policies to encourage the development and improve the performance of MSMEs.  One of which is to conduct a coaching and development program for MSME skills to focus on the intellectual capital owned by MSMEs in Buleleng Regency." (found on line 585)

  1. The conclusion section is adjusted as follows:

"Following the research that has been done and the results obtained, the implications of this research on improving the financial performance of MSMEs where it is necessary to maximize the potential of human resources owned by MSMEs to increase competitive advantage. The results of this study are expected to be a reference and consideration related to intellectual capital issues and speed of innovation and how these things can affect the improvement of financial performance by MSMEs. Through this research, MSMEs are expected to pay more attention to and utilize all human capital, structural capital, and consumer capital to improve performance, especially financial performance. To determine the best competitive strategy that MSMEs can carry out in Buleleng Regency, not only intellectual capital but the speed of innovation must be maximized. For the government, the results of this study can be taken into consideration by the Buleleng district government in setting policies to encourage the development and improve the performance of MSMEs, one of which is to conduct a coaching and development program for MSME skills with the target of focusing on the intellectual capital owned by MSMEs in Buleleng Regency (found on line 579)

The limitations of this study are that the variables used are only three independent variables: human, structural capital, and consumer capital, as well as the moderating variable, namely the speed of innovation, so that it still does not show a role in improving financial performance. They are representing MSMEs outside the Buleleng Regency area. It is hoped that further research will be able to develop variables and research sites supporting financial performance and use different analytical techniques to strengthen research results. (found on line 601)

  1. Introduction and discussion have been adapted to the research problem as follows:

“Both large, medium, and MSME companies in carrying out their operational activities strive to have the most important asset, namely human resources which have a major influence on the success of a business. However, in various studies conducted, it is still found that the frequent and serious obstacle faced by MSMEs is the low quality of human resources, not least what happened to MSME actors in Buleleng Regency. Based on a study by the Balitbang Buleleng, this was felt during the COVID-19 pandemic where several MSME actors experienced losses and even went bankrupt because they did not improve the quality of their business souls, such as not carrying out innovation to maintain their business development. In addition, low education and lack of mastery of science and technology make it difficult for MSME actors to accept and adapt to the renewal process. This is the reason why the research was conducted in the Buleleng district. In addition, this research was conducted because previous research only examined the problems faced by MSMEs in the field of intellectual capital, so the researchers added several aspects to create research novelties and the results of previous studies were still diverse. Furthermore, the difference between this study and previous research lies in the research location, research subjects, and analytical techniques used. Based on the background and problems raised, this study aims to determine 1) the effect of human capital, structural capital, and consumer capital on improving financial performance, and 2) the speed of innovation as a moderating variable (strengthening or weakening) to determine its effect on the relationship between capital human capital, structural capital and consumer capital to improve financial performance” (found on line 114)

“Intellectual capital consisting of human capital, structural capital, and consumer capital is needed by MSME actors. Especially in Buleleng district is facing competition and improving MSME financial performance, where things need to be considered regarding the guidance and development of human resources, organization, application of technology, and the application of marketing and promotion. Meanwhile, the speed of innovation does not always have a positive impact in order to improve financial performance, because some things such as innovation are conveyed to customers too quickly so that they are not always well-received” (found on line 568)

Round 2

Reviewer 2 Report

Good luck!